# State of Charge Estimation of a Composite Lithium-Based Battery Model Based on an Improved Extended Kalman Filter Algorithm

**Ning Ding [1],\*, Krishnamachar Prasad [1], Tek Tjing Lie [1] and Jinhui Cui [2]**

[1]  School of Engineering, Computer and Mathematical Sciences, Auckland University of Technology, Auckland 1010, New Zealand; krishnamachar.prasad@aut.ac.nz (K.P.); tek.lie@aut.ac.nz (T.T.L.)

[2]  Chongqing Datang International Wulong Hydropower Development Co., Ltd., Wulong 408500, Chongqing, China; cuijinhui@126.com

\*  Correspondence: nding@aut.ac.nz; Tel.: +64-021-0270-3708

**Abstract:** The battery State of Charge (SoC) estimation is one of the basic and significant functions for Battery Management System (BMS) in Electric Vehicles (EVs). The SoC is the key to interoperability of various modules and cannot be measured directly. An improved Extended Kalman Filter (iEKF) algorithm based on a composite battery model is proposed in this paper. The approach of the iEKF combines the open-circuit voltage (OCV) method, coulomb counting (Ah) method and EKF algorithm. The mathematical model of the iEKF is built and four groups of experiments are conducted based on LiFePO4 battery for offline parameter identification of the model. The iEKF is verified by real battery data. The simulation results with the proposed iEKF algorithm under both static and dynamic operation conditions show a considerable accuracy of SoC estimation.

**Keywords:** composite battery model; state of charge; improved extended Kalman filter; state of charge estimation

## 1. Introduction

In response to the issues of the energy crisis and environmental pollution, Electric Vehicles (EVs) are described as one of the most significant developments in the transport industry. The difficulties of the manufacturing technology and the assembly implementation of the battery seem to restrain the development of EVs. The battery as the main power source directly affects vehicle performance. The Battery Management System (BMS) is a significant issue to the battery utilization in the vehicle. The main functions of BMS include improvement of the effectiveness of the battery utilization, prevention of over-charging and over-discharging, prolonging the battery life and monitoring of the battery status. The high effectiveness of the BMS in EVs will mainly depend on accurate information exchange between the modules. The basic function of the monitor module in BMS is to capture and monitor the status of the battery and includes the state of charge (SoC), state of power (SoP), and state of health (SoH) [1]. The SoC of the battery is a crucial element to decide the battery operating condition. In terms of practical applications, the SoC is also the key factor to ensure the implementation of each module function in the BMS [2].

The use of different types of batteries in EVs started from the lead-acid battery and quickly progressed to nickel-metal hydride (NiMH) battery and eventually to lithium-ion and ternary lithium battery [3]. The working trend of the cell and battery pack in EVs is a dynamic and non-linear change, specifically in the lithium-based battery applications [4]. The value of SoC is not directly measurable. The SoC is affected by serval factors such as the battery terminal voltage, current and temperature.

Moreover, measurements of voltage, current, etc. often occur with inevitable noises that lead to errors. The issues of accurate estimation of SoC are being continuously discussed and improved [5].

The commonly used estimation methods of SoC mainly are based on Open Circuit Voltage (OCV) measurement, the coulomb counting method (Ah method), the method of Electrochemical Impedance Spectroscopy (EIS), the method based on neural network, the Kalman Filter (KF) based methods, etc. [6]. The OCV-based estimation, Ah method and EIS could be described as direct measurement methods. Taking into account the practicality, the advantage of direct measurement methods is its easy implementation and low cost. However, the values of OCV and the current are obtained through an open-loop estimator. The hysteresis phenomena and the sensor drift will lead to inevitable and cumulative errors, which will significantly impact the operation of the BMS and the use of battery itself [2]. The estimation approach of the neural network is a data-driven method. The main purpose of the data-driven method is to simulate the nonlinear and dynamic characteristics of the battery. This method requires large Random Access Memory (RAM) storage to train the system for the learning process of the neural network [7]. Other machine learning methods such as the fuzzy logic, support vector machine and genetic algorithm have been extensively researched for battery SoC estimation [8–11]. The machine learning method improves the intelligence of the system but the results are difficult to interpret and hence is not convenient in real practice. The need for appropriate storage requirement appears in the look-up table method of the SoC estimation. The look-up table sourced from the relationship curve of SoC-OCV is inadvisable for online application. A regular recalibration is required to update the table information [12].

Another significant approach of SoC estimation is the model-based method, which is robust and is based on a close-loop-feedback system. The standard KF method is a well-known approach to estimate the internal state of a dynamic linear system [13]. To precisely describe the nonlinear working state of the battery, several KF relevant approaches have been used to obtain SoC estimation [14]. The extended KF (EKF) method mathematically transforms the standard (linear matrix) KF to fit a nonlinear system. Besides, several EKF-based methods, such as lazy EKF and robust EKF, are proposed to estimate the SoC of the battery and in different scenarios for vehicle onboard battery and microgrid energy storage units [15,16]. The Dual EKF (DEKF) adopted in [17,18] to estimate the SoC of the battery and the parameter of the model shows faster convergence in shorter calculation time. The unscented KF (UKF) transforms the nonlinear models by linear interpolation. Sun et al. [19] discussed an adaptive UKF (AUKF) algorithm for SoC estimation which shows higher accuracy than the EKF-based estimator. The main issue of the UKF based methods is using the Unscented Transform to deal with the nonlinear transfer of the mean and covariance. Compared to the EKF, the UKF method approximates the probability density distribution of the nonlinear function instead of the nonlinear function approximation. Therefore, the UKF methods are more accurate [20–22]. However, the UKF-based estimator seems more applicable to the situation where the system is less nonlinear. The iterative update of the covariance in the UKF-based algorithm is prone to negative definite matrix in a highly nonlinear system. In addition to the methods related to KF, other model-based methods are reported in literature. For example, the Luenberger observer and sliding mode observer showed an accurate SoC estimation but is very complex [23].

The main objective of this paper is to design an appropriate KF-based method for SoC estimation with considerable accuracy and simplicity. The battery SoC estimation is the parameter identification based on the battery model. In this paper we employ a novel composite battery model based on the electrochemical models [24]. The offline identification of parameters is first done using experimental data. With these offline parameters, a battery model is subsequently built using MATLAB simulation. This model will include the effects of temperature, charge/discharge rate, direct current resistance, etc. Using this battery model, we implement the iEKF algorithm to estimate the SoC in MATLAB Simulink. The iEKF method combines the OCV, Ah and EKF methods: the OCV-SoC function in our work provide an initial value; Ah counting module online identifies the parameters of the battery model; the errors in the OCV and Ah estimation are corrected by the EKF algorithm. Under both static

and dynamic operating conditions, it is shown that the iEKF algorithm results in better accuracy of the SoC estimation than other EKF based methods. The development of a composite battery model is another novelty of this paper.

## 2. The Battery Model

The State of Charge (SoC) is used to identify the remaining capacity status of the battery. The value of SoC cannot be directly measured but obtained by analyzing external characteristics. Different battery manufactures in the market have a different definition of the SoC. Equation (1) gives a numerical definition of SoC.

$$\text{SoC} = C_Q/C_I \tag{1}$$

where the $C_Q$ is the remaining capacity, and the $C_I$ is the rated capacity when the battery is discharged with a constant current $I$.

The process of using the model-based method to estimate SoC could be considered as the dynamic identification for the parameters representing the characteristics of the battery. The choice of the models will determine the mathematical relationship of the different parameters. The equations of the state in the algorithms will also vary from the different model selections. Figure 1 shows a logical structure of the battery model build-up.

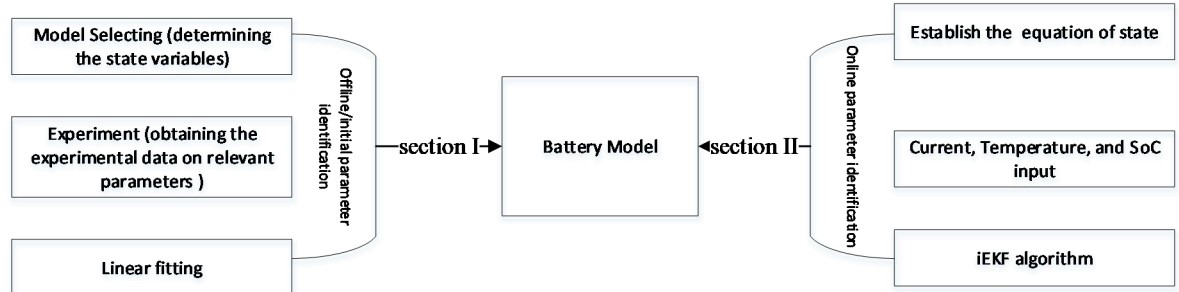

**Figure 1.** The logical structure of the battery model build-up.

The modeling steps are described as follows:

1.  Selecting the battery model (A novel composite electrochemical model).
2.  Determining the different parameters to be identified based on the selected model.
3.  Performing a series of characteristics tests on the battery.
4.  Linear fitting of the experimental data (offline identification).
5.  Building the model in MATLAB.
6.  Obtaining the equation of state.
7.  Developing the online estimation algorithm (iEKF).
8.  Inputting the current, temperature and SoC (simulations in static and dynamic conditions).

As shown in Figure 1, the battery model is constructed in two sections: offline parameter identification in section I and online parameter identification in section II. Steps 1 to 6 are the preparation steps for the battery model establishment. They will seriously influence the accuracy of the online estimation based on the iEKF algorithm in step 7. The series of experiments in step 3 refers to four different battery tests (see Sections 3.1 and 3.4). The experiments reflect the external characteristics of the battery and relate to the factors affecting the SoC estimation. The main factors affecting the battery capacity are the charge/discharge rate, temperature, the charge/discharge cycle (battery aging) and self-discharge of the battery [25]. Taking these factors into account, SoC ($Q_{tI}$) definition is amended as [26,27].

$$Q_{tI} = \eta Q_{ti} = \eta \int_0^t I d\tau \tag{2}$$

where $t$ is the charge/discharge duration time, $Q_{ti}$ is the total quantity of electric charge/discharge during the period $t$, $I$ is the current over a time interval 0 to $t$ and $\eta$ is the efficiency coefficient which includes both the charge/discharge rate ($\eta_i$) and the temperature influence coefficient ($\eta_T$).

*The Composite Battery Model*

For the composite battery model, the SoC is considered as a state variable x in the system, $x_k$ is the number of the state vector. The output $y_k$ is the voltage of the battery model. The composite model is built based on three different electrochemical models as follows [28,29]:

Shepherd model:

$$y_k = E_0 - Ri_k - K_1/x_k \tag{3}$$

Unnewehr universal model:

$$y_k = E_0 - Ri_k - K_2 \cdot x_k \tag{4}$$

Nernst model:

$$y_k = E_0 - Ri_k + K_3 \cdot \ln(x_k) + K_4 \cdot \ln(1 - x_k) \tag{5}$$

where $E_0$ is the OCV when the battery is fully charged and $R$ is the internal resistance which will change with different charge/discharge status and SoC, $i_k$ is the instantaneous current at time $k$ (negative when the battery is charging and positive when discharging). $K_1$ to $K_4$ are the matching parameters to be identified through battery experiments. The state equation based on the composite battery model is described as follows:

$$x_{k+1} = x_k - (\eta_i \Delta t / \eta_T Q_n) \cdot i_k \tag{6}$$

While the output equation is

$$y_k = K_0 - Ri_k - K_1/x_k - K_2 \cdot x_k + K_3 \cdot \ln(x_k) + K_4 \cdot \ln(1 - x_k) \tag{7}$$

where $K_0$ is the OCV of the fully charged battery and has the same physical meaning as $E_0$. However, $E_0$ in Equations (3)–(5) is the actual measured value while $K_0$ is obtained by the identification based on OCV-SoC experimental data (see in Section 3.4). Equations (3)–(5) are combined in Equation (7). The electrochemical models of Equations (3)–(5) reflect the relationship between the terminal voltage and the SoC ($x_k$). $R$ is the internal resistance (Ohmic resistance) and changes with the charging/discharging state of the battery ($Ri_k$). $K_1/x_k$ from Equation (3) and $K_2 \cdot x_k$ from Equation (4) reflect the polarization resistances of the battery. $K_3 \cdot \ln x_k$ and $K_4 \cdot \ln(1 - x_k)$ from Equation (5) represent the influence of the internal temperature and material activity during the electrochemical reaction of the battery, respectively.

The SoC estimation is not directly considered in most of the papers discussing the battery models. On the other hand, the accuracy of the battery model significantly affects the estimation of the SoC of the battery [30]. Justifying accurate battery models mainly depends on the dynamic tracking of the battery terminal voltage and the identification of parameters. Many bench tests are used to verify the accuracy of the battery models, such as HPPC test, pulsed charge/discharge cycles and dynamic stress test [31,32]. At the same time, the experimental data used to verify the battery model is also applied to design the SoC estimators.

In terms of the electrochemical model, various chemical reactions are carefully considered. They include the reaction occurring at the anode and cathode of the battery and the electrolyte ion transfer process. Equations (3)–(5) are simplified electrochemical models based on empirical modeling method and have the advantages of simple expression and computational efficiency. However, there are some drawbacks when using models in Equations (3)–(5) alone or their correction models such as the new Electrochemical Polarization (EP) model [22,32]. Compared to the equivalent circuit models (ECMs) [33,34], the composite model described in our work shows sufficient accuracy and short execution time. The composite battery model in this paper uses the calculation results of Equation (7)

to replace the direct measurement of OCV, which is known to be difficult [35] but still used in other electrochemical models and ECMs. The advantage of linear parameters in our model reduces the difficulty of parameter identification. With less complexity, our composite battery model minimizes the number of the parameters to be identified while fully considering the influencing factors by combing $K_1$ to $K_4$.

## 3. The Experiments and Offline Parameter Identification

The mathematical parameters ($\eta_i$, $\eta_T$, $R$, $K_0$, $K_1$, $K_2$, $K_3$, $K_4$) of the composite battery model shown in Equations (6) and (7) require a validation. The initial identification of the parameters is an offline and static estimation, obtained by polynomial curve fitting of the experimental data (see Figures 2–8). The charge/discharge rate factor $\eta_i$ was obtained by the charge and discharge test. Similarly, the temperature influence coefficient $\eta_T$ was obtained from temperature characteristic test. The other parameters ($R$, $K_0$, $K_1$, $K_2$, $K_3$, $K_4$) were obtained through a Hybrid Pulse Power Characterization (HPPC) test and OCV-SoC test [36]. All the tests were conducted on a LiFePO$_4$ battery (single cell) with 206 Ah rated capacity and 3.2 V rated voltage.

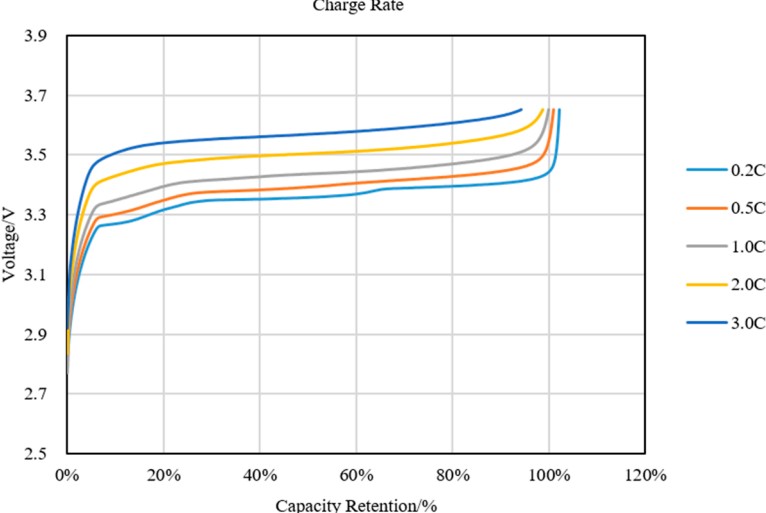

**Figure 2.** Capacity retention curves at different charge rates.

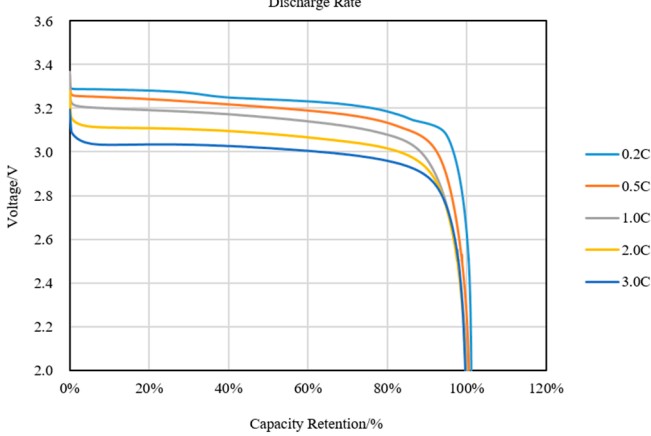

**Figure 3.** Capacity retention curves at different discharge rates.

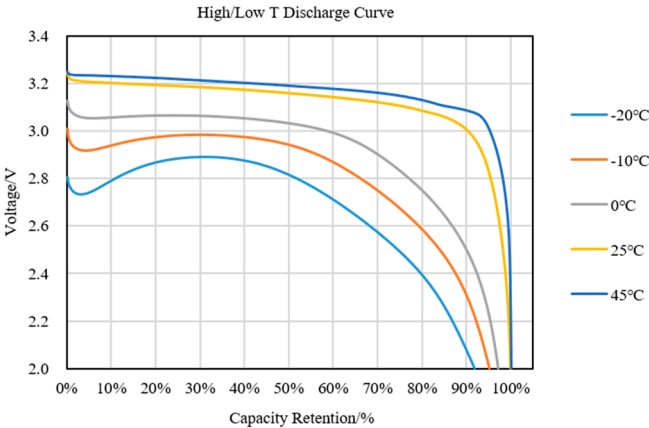

**Figure 4.** Capacity retention curves at a different temperatures.

(**a**)

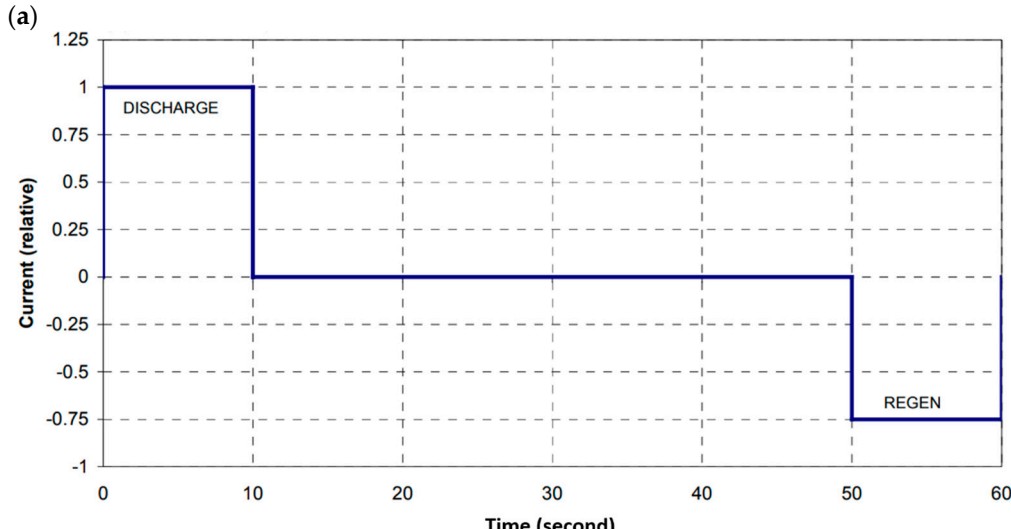

(**b**)

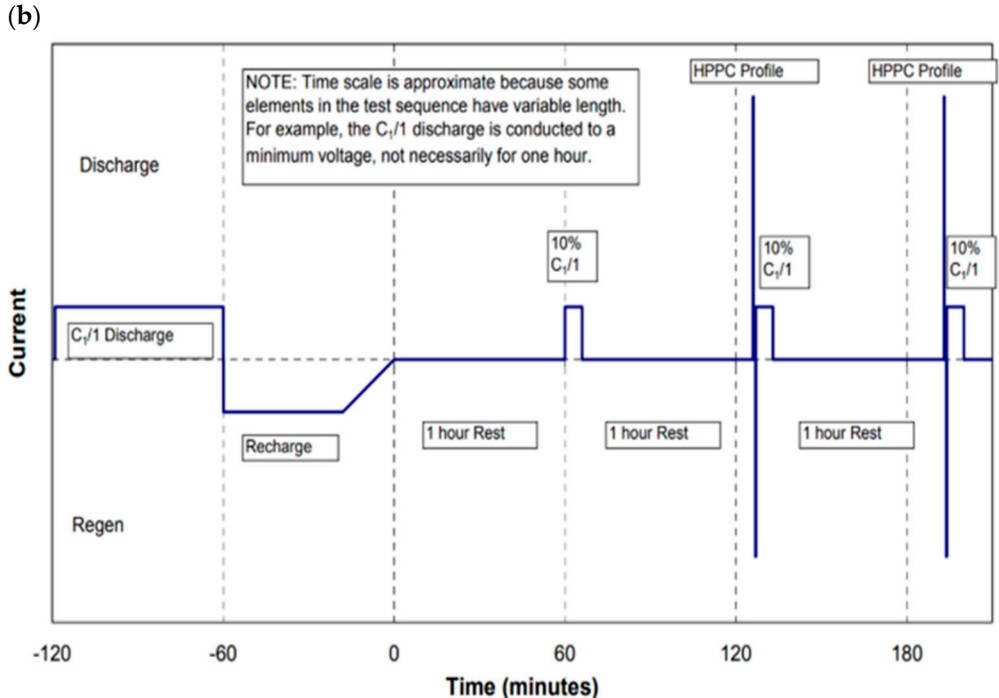

**Figure 5.** *Cont*.

**(c)**

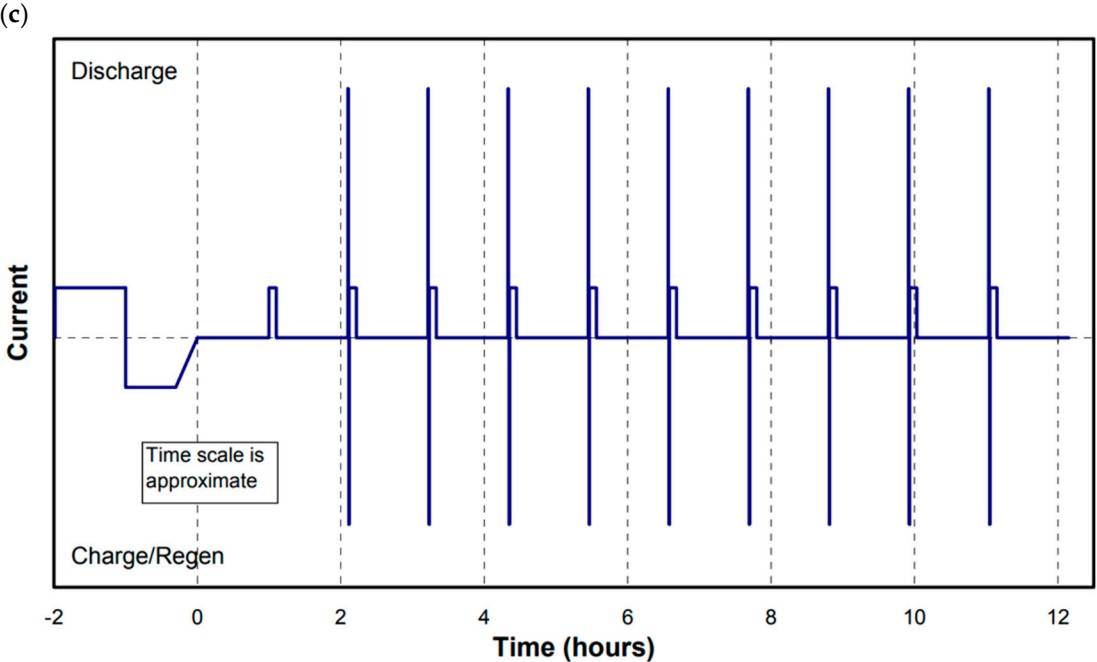

**Figure 5.** The HPPC test [36]. (**a**) the HPPC test profile, (**b**) the start of HPPC test sequence, (**c**) the complete HPPC sequence.

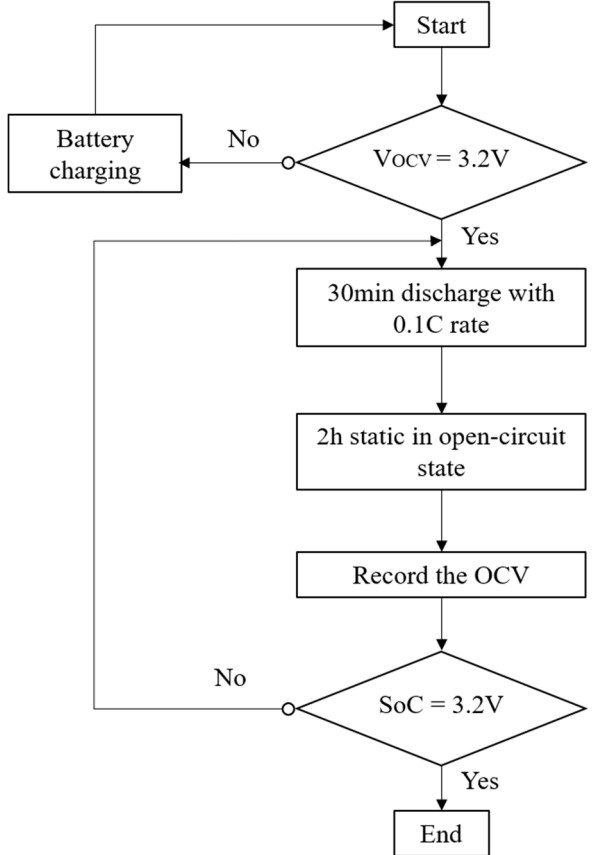

**Figure 6.** OCV-SoC test process.

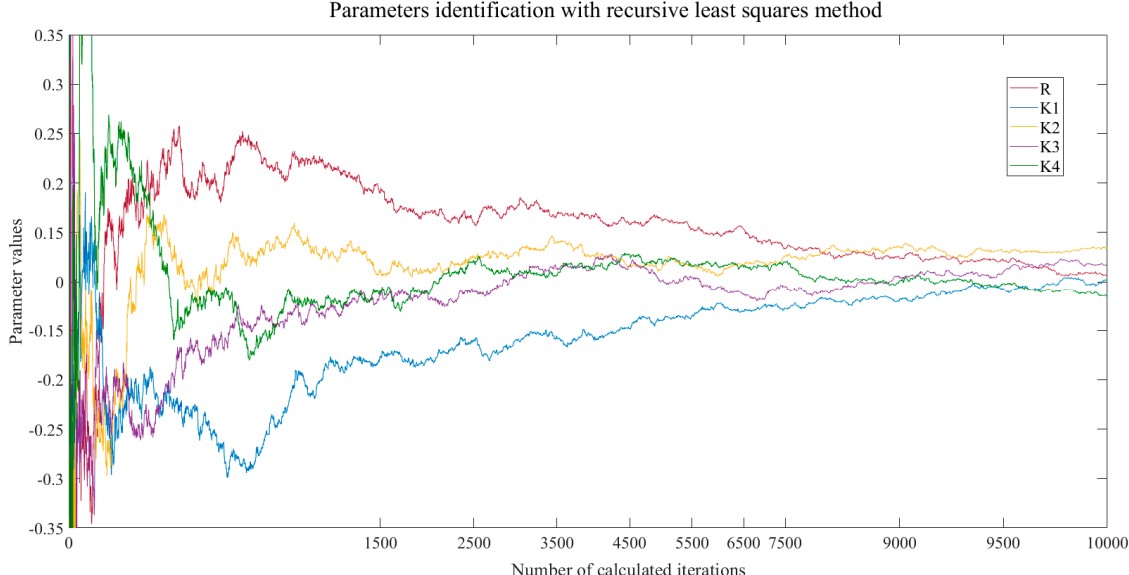

**Figure 7.** Parameters identification results in the composite model.

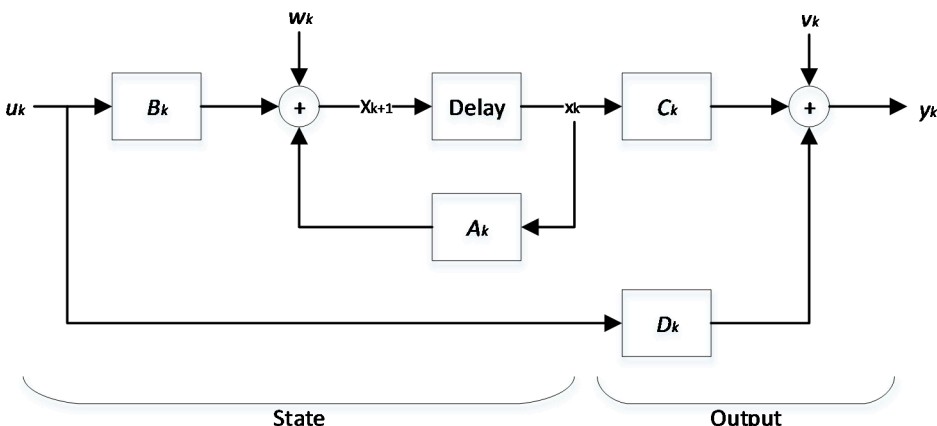

**Figure 8.** The state-space model of discrete simple KF.

### 3.1. Charge and Discharge Rate Test

The main purpose of this test is determining the influence from different charging/discharging rates on the actual capacity of the battery. According to the definition of SoC shown in Equation (2), the difference between the rated capacity and the measured capacity is caused by the variable parameter of $\eta_i$. The determination of $\eta_i$ is based on the experimental data to establish a linear fit relationship.

Five groups of different rates (0.2 C, 0.5 C, 1 C, 2 C and 3 C) with a constant current at room temperature (25 °C) were conducted in charging and discharging tests, respectively. There was a 1 h resting time before starting the discharging test after the charging test was completed. Tables 1 and 2 show a specific data point of both capacity and energy retention for charging and discharging tests, respectively. The curves of the capacity retention rate for the charge and discharge rate tests for the LiFePO$_4$ battery are shown in Figures 2 and 3, respectively. From Figures 2 and 3, it is clear that the actual capacity of the battery will decrease as the charge/discharge rate increases.

**Table 1.** The capacity and energy retention rate with specific charge rate.

| Charge Rate Test | 0.2 C | 0.5 C | 1.0 C | 2.0 C | 3.0 C |
|---|---|---|---|---|---|
| Capacity/Ah | 211.45 | 208.95 | 206.88 | 204.07 | 194.76 |
| Capacity retention rate/% | 102.21 | 101.00 | 100.00 | 98.64 | 94.14 |
| Energy/Wh | 707.76 | 706.36 | 708.04 | 712.64 | 692.15 |
| Energy retention rate/% | 99.96 | 99.76 | 100.00 | 100.65 | 97.76 |

**Table 2.** The capacity and energy retention rate with specific discharge rate.

| Charge Rate Test | 0.2 C | 0.5 C | 1.0 C | 2.0 C | 3.0 C |
|---|---|---|---|---|---|
| Capacity/Ah | 214.31 | 212.94 | 211.84 | 211.28 | 211.08 |
| Capacity retention rate/% | 101.16 | 100.52 | 100.00 | 99.74 | 99.64 |
| Energy/Wh | 687.83 | 672.22 | 657.33 | 641.41 | 628.45 |
| Energy retention rate/% | 104.64 | 102.26 | 100.00 | 97.58 | 95.61 |

### 3.2. Temperature Characteristics Test

Similar to the charge/discharge rate, the battery shows nonlinear behavior with ambient temperature changes. The working temperature of the lithium-based battery pack in actual situations is not a constant. Therefore, a temperature coefficient ($\eta_T$) has to be introduced to represent the impact of temperature on the SoC estimation.

The test involved using the 1 C rate to discharge a fully charged battery under different (constant) temperature conditions ($-20\,°C, -10\,°C, 0\,°C, 25\,°C$ and $45\,°C$). Table 3 shows a specific data point in both the capacity and energy retention rates at specific temperatures. The change curves of capacity retention rate for the LiFePO$_4$ battery at different temperature are shown in Figure 4.

**Table 3.** The capacity and energy retention rate with specific temperature condition.

| Temperature Characteristic Test | −20 °C | −10 °C | 0 °C | 25 °C | 45 °C |
|---|---|---|---|---|---|
| Capacity/Ah | 194.18 | 201.14 | 205.49 | 211.27 | 211.35 |
| Capacity retention rate/% | 91.91 | 95.20 | 97.26 | 100.00 | 100.00 |
| Energy/Wh | 523.18 | 566.18 | 600.52 | 657.80 | 668.46 |
| Energy retention rate/% | 79.54 | 86.07 | 91.29 | 100.00 | 101.62 |

The increase in temperature reduces the overpotential, which will cause an increase in the activity of the chemicals inside the battery. Thus, the discharge capacity of the battery increases at (constant) higher temperature conditions. The experimental results shown in Figure 4 are consistent with the theoretical analysis—the capacity of the LiFePO$_4$ battery decreases with the decreasing temperature [37].

### 3.3. Hybrid Pulse Power Characterization (HPPC) Test

The HPPC test is one of the most common approaches to offline parameter identification. This paper adopts the HPPC from Freedom CAR Battery Test Manual (published in 2003) to initially identify the parameters of direct current resistance (DCR) [36]. The DCR, which includes both Ohmic resistance and the polarization resistance, is dynamic. The HPPC test is used to obtain DCR parameters as a function of SoC from the voltage response values. The series of pulse experiments of HPPC is to reliably establish cell voltage response time constants during "discharge", "rest" and "regen" regions. The standard process of HPPC test is described in Figure 5, where C$_1$/1 is 1C charge/discharge rate.

The HPPC test includes nine single repeated sequences (Figure 5c). Each sequence is separated by 10% SoC or depth of discharge (DOD) with constant current discharge segments from 90% to 10% (Figure 5b). One HPPC profile includes 10 s discharge, 40 s rest and 10 s charge (Figure 5a). There is one hour resting time between each sequence (Figure 5b). It should be noted that the spikes shown in

Figure 5b,c are basically the HPPC profile of Figure 5a. This test uses 4 C discharge and 3 C charge. The data collection interval of the HPPC test is 0.2 s. The HPPC experimental results are shown in Table 4 and include the voltage and the DCR. $V_{before}$ and $V_{after}$ are the measured voltage values before and after charging/discharge test, respectively. DCR values are rounded to two decimal places.

**Table 4.** The HPPC test charging/discharging power and DCR.

| SoC | 206Ah 4C Discharge/3C Charge | | | | | |
| | Discharge | | | Charge | | |
| | $V_{before}$ (mV) | $V_{after}$ (mV) | DCR (mΩ) | $V_{before}$ (mV) | $V_{after}$ (mV) | DCR (mΩ) |
| --- | --- | --- | --- | --- | --- | --- |
| 90% | 3328.30 | 2910.70 | 0.45 | 3277.20 | 3613.20 | 0.49 |
| 80% | 3327.40 | 2893.70 | 0.47 | 3267.20 | 3612.90 | 0.50 |
| 70% | 3327.40 | 2881.60 | 0.48 | 3258.90 | 3610.10 | 0.51 |
| 60% | 3296.70 | 2857.40 | 0.48 | 3244.90 | 3586.90 | 0.50 |
| 50% | 3289.00 | 2829.80 | 0.50 | 3230.40 | 3580.40 | 0.51 |
| 40% | 3288.00 | 2805.90 | 0.52 | 3217.30 | 3577.30 | 0.52 |
| 30% | 3284.30 | 2775.60 | 0.55 | 3201.50 | 3570.10 | 0.53 |
| 20% | 3254.50 | 2707.40 | 0.59 | 3169.90 | 3539.40 | 0.53 |
| 10% | 3209.60 | 2422.10 | 0.86 | 3110.40 | 3497.00 | 0.56 |

### 3.4. OCV-SoC Test

The open-circuit voltage (OCV) test is significant and necessary for the estimation of $K_0$. The operating characteristics of the battery show that a proportional relationship exists between OCV and SoC. The OCV is roughly regarded as a linearized function of SoC in a simplified system [38]. For example, the OCV rises with the increase in SoC. The current SoC of the battery can be calculated through a model relationship between OCV and SoC. The experimental steps used to obtain an approximate OCV value are shown in Figure 6.

### 3.5. Offline Parameter Identification

The data shown in the various tests described in Sections 3.1–3.4 was obtained by real experimental bench based on LiFePO$_4$ battery (206 Ah, 3.2 V). A convincing offline parameters identification was required to achieve a reliable mathematical battery model for the simulation of KF. Eight parameters ($\eta_i$, $\eta_T$, $R$, $K_0$, $K_1$, $K_2$, $K_3$, $K_4$) were identified by the approach of linear fitting and recursive least squares (RLS) in MATLAB. The experimental data was input into MATLAB in the form of different sets of data points. In terms of $\eta_i$, relationship between the actual capacity and the charge/discharge rate were quantified by linear fitting functions in MATLAB. A second-order polynomial linear fitting equation for $\eta_i$ was obtained by the use of the polyfit function:

$$\text{Polyfit}\ (i,\ \eta_i,\ 2)$$
$$\eta_i = 15873/\left(5.47i^2 - 156.8i + 16301\right) \tag{8a}$$

where $i$ represents the charging-discharging current in A. Similarly, the polynomial curve of the temperature coefficient ($\eta_T$) was obtained by MATLAB polyfit function, as follows:

$$\text{Polyfit}\ (T,\ \eta_T,\ 1)$$
$$\eta_T = 0.55T + 76.83 \tag{8b}$$

where $T$ is the actual temperature of the battery.

The rest of parameters ($K_0$, $R$, $K_1$, $K_2$, $K_3$, $K_4$) were obtained by the method of RLS. Figure 7 shows that the vales of parameters begin to converge after 1500 iterations. The values of the parameters were as follows: $R = 0.0048$, $K_1 = -0.000268$, $K_2 = 0.1495$, $K_3 = 0.111$ and $K_4 = -0.01955$. The value of $K_0$ was 3.191 and is not shown in the figure. The accuracy in the estimation of parameters was high,

of the order of ±0.3%. The mathematical model based on linear fitting from real experimental data is therefore reliable for SoC estimation using iEKF.

## 4. The SoC Estimation Based on an Improved EKF Algorithm

The crucial objective of addressing the filtering problem is filtering any noise from the observed noisy signal. Taking specific situations into account such as signal processing, target tracking and control system, the solution for filtering problem can be generally transformed into a statistical estimation of the system status. Based on the theory of time domain state space, the KF algorithm achieves an estimation of the state of the system by recursive iteration.

### 4.1. Analysis of the KF and EKF Algorithm

The standard KF is mainly used in a linear dynamic system to estimate the unknown variable which cannot be directly measured. The optimal estimation is the core of the KF algorithm, which is based on the prediction estimation and algorithmic amendment. The processing objects of the KF algorithm include the real system and the system model [39]. The real system includes the measurable input $u_k$, the real output $y_k$ and the unmeasurable state $x_k$. The system model includes the same input $u_k$, the known state $x_k$ and the output $y_x$ based on the specific battery model. The optimal estimation is obtained through a comparison between the $y_k$ and $y_x$ to amend the prediction estimation. The state variable $x_k$ of the system model is closer to the real value of $y_k$. The state-space system model of the discrete-time standard KF is expressed as follows:

$$\begin{cases} state\ formula:\ x_{k+1} = A_k x_k + B_k u_k + w_k \\ output\ formula:\ y_k = C_k x_k + D_k u_k + v_k \end{cases} \tag{9}$$

where $k$ is the discrete time point, $x_k$, $u_k$ and $y_k$ are the state variables of the input and the output of the system, respectively. $w_k$ is the process noise variable which is used to describe the superimposed noise and error during state transition. $v_k$ is the measurement noise variable which is used to describe the generated noise and error when the input is measured. $A_k$, $B_k$, $C_k$ and $D_k$ are the equation matching coefficients reflecting the dynamic characteristics of the system. Figure 8 shows the state-space model of the discrete simple KF.

Two different estimates of the state variable $x_k$ and the mean square error of estimation $P_k$ are made at each sampling interval. For example, the first-time predictive estimate of $x_k^-$ is obtained by the iterative recursion using the state equations based on $x_{k-1}^+$. The predictive estimates of $x_k^-$ and $P_k^-$ are completed before the $y_k$ measurement. The calculation of the optimal estimates of $x_k^+$ and $P_k^+$ start after the measurement of $y_k$ is processed. To obtain the optimal estimation of $x_k^+$ and $P_k^+$, the predictive estimates of $x_k^-$ and $P_k^-$ will be amended after the calculation of $y_k$.

The processing steps of standard KF algorithm are as follows:

1. Initial value of $x_0^+$ and $P_0^+$:

$$x_0^+ = \mathrm{E}[x_0] \tag{10}$$

$$P_0^+ = E\left[\left(x_0 - x_0^+\right)\left(x_0 - x_0^+\right)^T\right] \tag{11}$$

2. Predictive estimate of the $x_k^-$ and $P_k^-$:

$$x_k^- = A_{k-1} x_{k-1}^+ + B_{k-1} u_{k-1} \tag{12}$$

$$P_k^- = A_{k-1} P_{k-1}^+ A_{k-1}^T + D_w \tag{13}$$

3. KF gain $L_k$ (weighting coefficient matrix)

$$L_k = P_k^- C_k^T \left(C_k P_k^- C_k^T + D_v\right)^{-1} \tag{14}$$

4. Optimal estimate of the $x_k^-$ and $P_k^-$:

$$x_k^+ = x_k^- + L_k(Y_k - y_k) \tag{15}$$

$$P_k^- = (1 - L_k C_k) P_k^- \tag{16}$$

where $D_w$ and $D_v$ in Equations (14) and (15) are the covariance of the process noise $w_k$ and the measurement noise $v_k$, respectively.

The standard KF algorithm shows the advanced estimation of the linear dynamic system. In terms of the nonlinear dynamic system such as the battery pack of EVs, the EKF algorithm linearly transforms the nonlinear system through an extended state-space model, then uses the iterative calculation of the standard EK algorithm to obtain the optimal estimation [40]. The state-space system model of the EKF is expressed as:

$$\begin{cases} state\ formula:\ x_{k+1} = f(x_k, u_k) + w_k \\ output\ formula:\ y_k = g(x_k, u_k) + v_k \end{cases} \tag{17}$$

where $f(x_k, u_k)$ and $g(x_k, u_k)$ are the state transfer function and the measurement function of the nonlinear system, respectively. The nonlinear discrete-time state-space model of the EKF algorithm is shown in Figure 9.

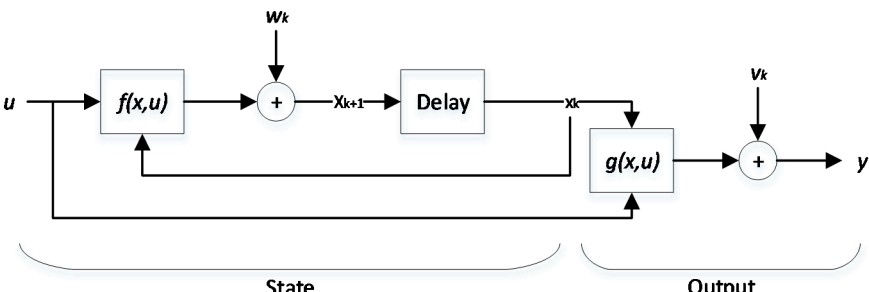

**Figure 9.** The state-space model of EKF.

Compared to the standard KF algorithm, the state-space model of EKF algorithm is different. However, the algorithm implementation is similar to the standard KF, which mainly includes the initialization, the predictive estimate and the optimal estimate. The $A_{k-1} x_{k-1}^+ + B_{k-1} u_{k-1}$ in predictive estimate of standard KF is replaced by $\left(x_k^-, u_k\right)$, while $g\left(x_k^-, u_k\right)$ substitutes $C_k x_k + D_k u_k$ in the optimal estimate. The calculation process of EKF is shown in Figure 10.

### 4.2. The SoC Estimation Model Corrected by EKF Based on Composite Model

The complexity of the SoC estimation increases because of the internal nonlinearization of the battery and the significant impact of external conditions. In addition, the working process of the battery for EVs involves large current changes and a single and regular approach hardly achieves an online accurate SoC estimation.

Taking the estimate accuracy and the calculation cost into consideration, an iEKF algorithm is proposed in this paper. The iEKF algorithm combines the OCV method, Ah method and the EKF algorithm. Based on the actual experimental battery data and the initial/offline parameter extraction using MATLAB, the algorithm implementation of the iEKF is shown in Figure 11.

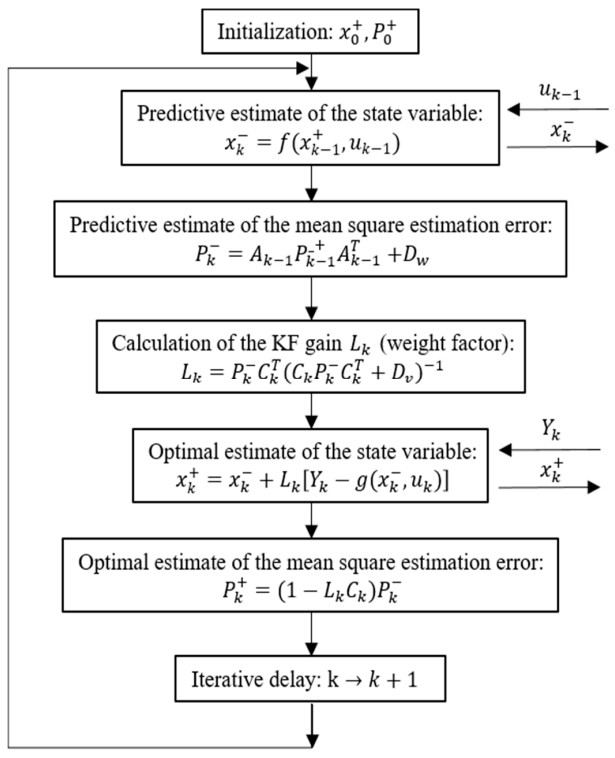

**Figure 10.** The calculation process of EKF.

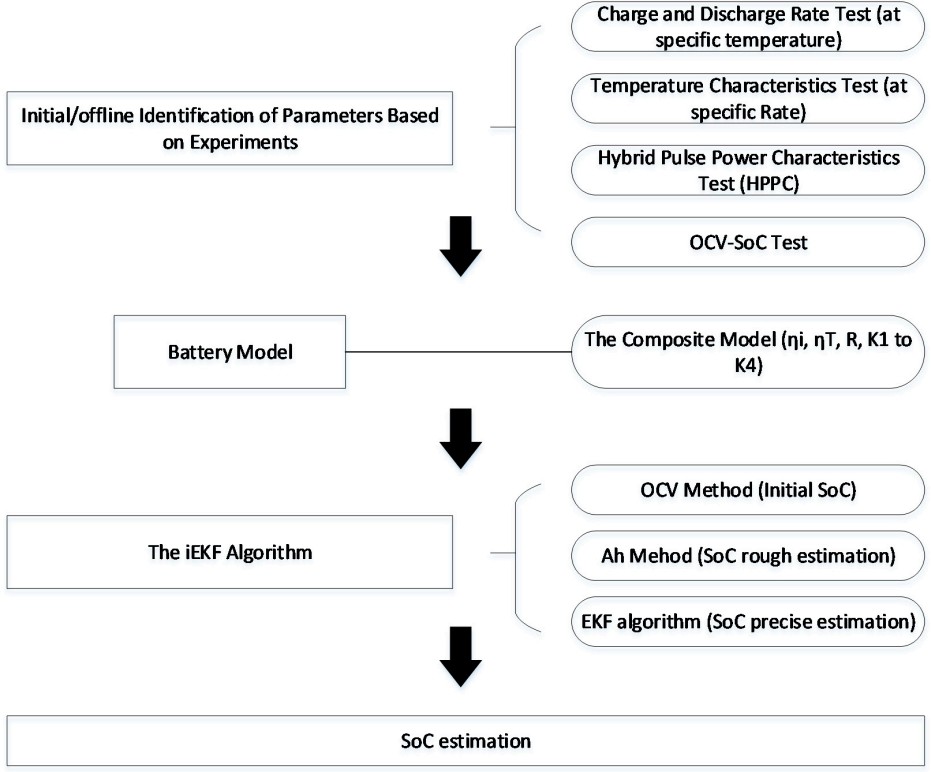

**Figure 11.** The iEKF algorithm implementation.

The main function of OCV method is to provide a relatively accurate initial estimation of the SoC. OCV-SoC curve obtained from the OCV-SoC test (see Section 3.4) is used to provide an initialized SoC estimation for Ah method and EKF algorithm. A piecewise function is used in this paper for curve

fitting the experimental data in OCV-SoC test. The mathematical expression of OCV-SoC relation is as follows ($x$ is the measured OCV):

$$\begin{cases} x > 3335, \ SoC = 1 \\ 3327 \leq X \leq 3335, SoC = 1 - \frac{8.4745x^2 - 56710x + 94872550}{14512} \\ 3297 < x < 3327, SoC = 1 - \frac{-0.26997x^2 + 1740x - 2798620}{14512} \\ 3283 \leq x \leq 3297, SoC = 1 - \frac{7.1273x^2 - 47222x + 78219350}{14512} \\ 3179 < x < 3283, SoC = 1 - \frac{-0.2017x^2 + 1256.2x - 1941900}{14512} \\ 2828 \leq x \leq 3179, \ SoC = 1 - \frac{-0.00735x^2 + 40.533x - 41359}{14512} \\ x < 2828, SoC = 0 \end{cases} \tag{18}$$

The Ah method quantifies the external influence factors which mainly refer to the charging/discharging rate and the temperature. The principle of Ah method is shown in (20).

$$SoC_{k+1} = SoC_k - \frac{1}{Q_n} \int_k^{k+1} \frac{\eta_i}{\eta_T} idt \tag{19}$$

where $i$ is positive when discharging, but negative when charging. The coefficient of charging/discharging rate ($\eta_i$) and temperature ($\eta_T$) are obtained by Equations (8a), (8b) and (9), respectively.

In terms of the implementation of EKF algorithm, the composite model described in Equations (3)–(5) is adopted in this paper. The state-space model of EKF algorithm based on the composite battery model uses Equation (6) as the state equation and the Equation (7) as the output equation. The EKF is expected to show a strong algorithm correction ability [41,42]. The problem of inaccurate initialized estimation of OCV method and the current accumulated error in Ah method will be addressed by using EKF.

The implementation of EKF algorithm in this paper is as follows:

1. Model establishment: Use Equations (6) and (7).
2. Determination of system parameters:

$$A_{k-1} = \left. \frac{\partial f(x_{k-1}, u_{k-1})}{\partial x_{k-1}} \right|_{x_{k-1} = x_{k-1}^+} = 1 \tag{20}$$

$$C_k = \left. \frac{\partial y_k}{\partial x_k} \right|_{x_k = x_k^-} = K_1 / \left( x_k^- \right)^2 - K_2 + K_3 / x_k^- - K_4 / \left( 1 - x_k^- \right) \tag{21}$$

3. Initialization of the state variable and the covariance.

$$x_0^+ = SoC_0, \ P_0^+ = var(x_0) \tag{22}$$

4. Iterative calculation of the EKF.

$$\begin{cases} x_k^- = x_{k-1}^+ - \left( \frac{\eta_i \Delta t}{\eta_T Q_n} \right) i_{k-1} \\ y_k = K_0 - Ri_k - K_1 / x_k^- - K_2 x_k^- + \\ \quad K_3 \ln\left(x_k^-\right) + K_4 \ln\left(1 - x_k^-\right) \\ P_k^- = A_{k-1} P_{k-1}^+ A_{k-1}^T + D_w \\ L_k = \frac{P_k^- C_k^T}{C_k P_k^- C_k^T + D_v} \\ x_k^+ = x_k^- + L_k(Y_k - y_k) \\ P_k^+ = (1 - L_k C_k) P_k^- \\ k = 1, 2, 3 \dots \end{cases} \tag{23}$$

$A_{k-1}$ and $C_k$ are defined in Step 2 by using Equations (6), (7) and (10). The SoC$_0$ is calculated based on the remaining charge (after charging/discharging) in the previous state and the OCV in the current state. $P_0^+$, $D_w$ and $D_v$ relate to the performance of the battery and the data collection system. In order to update the status of the system, the sampling frequency is set in the Simulink equal to 2.5 times the bandwidth of the sampled signal as per the "Nyquist-Shannon sampling" criterion [43].

## 5. The Simulation Validation of the Improved EKF Algorithm

### 5.1. The Validation of the Improved EKF Algorithm

In order to verify the actual effect of the iEKF, the model described in Section 4 has been simulated in MATLAB/Simulink. The inputs of the model, current and temperature, are set by constant values (1 C, 2 °C) which refer to the static operation condition. The initial guess for SoC is set at 70% based on existing published work [44–47].

### 5.2. The Simulation Results

Figure 12 shows the comparison between standard SoC, the SoC estimated by Ah counting model and the SoC estimation results of iEKF method. The standard SoC was calculated values based on the charging/discharging experimental data [48]. The fit between the standard SoC and the iEKF estimation gets better with increasing simulation time.

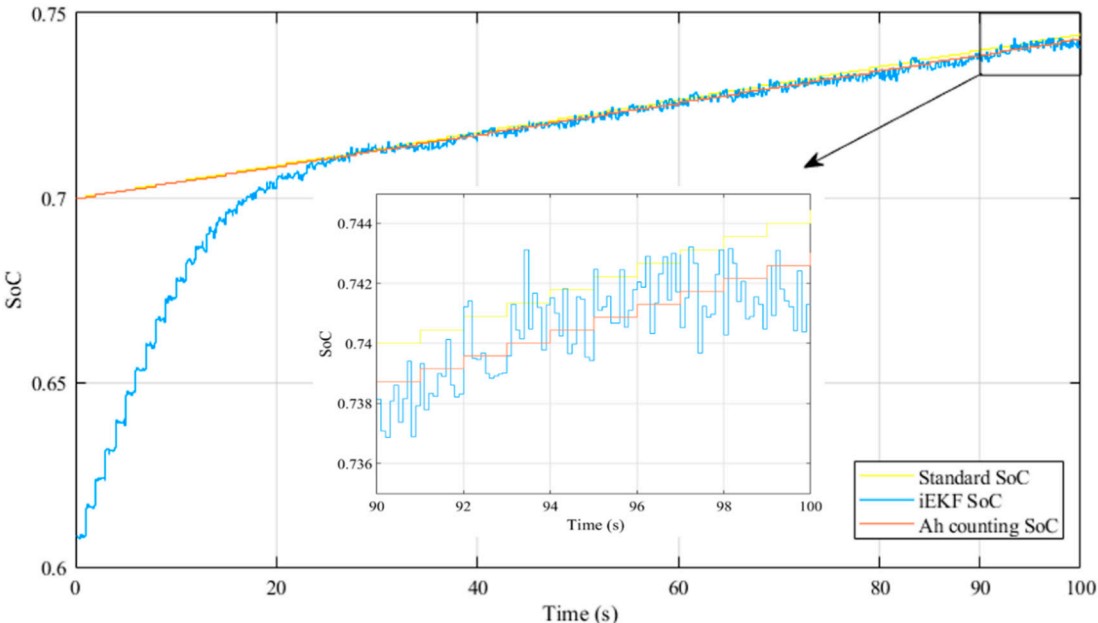

**Figure 12.** Comparison SOCs under static operation conditions.

Figure 13 shows the error covariance of the composite SoC model with and without the iEKF under static operation condition. The error with the use of iEKF is very small reflecting the effectiveness of the improved EKF method. The error covariance with the iEKF under static operation condition was of the order of $10^{-7}$.

The MATLAB inputs of dynamic current efficiency and temperature are shown in Figure 14a,b, respectively. Figure 15 compares the SoC estimation error under dynamic operation condition of the composite battery model with and without the iEKF. Figure 15a gives the calculation results over a time of 1000 s while Figure 15b shows local calculation results from 0 to 100 s. The estimation error using iEKF is insignificant (of the order of $10^{-6}$), thereby providing credibility to the iEKF method. The error covariance with the iEKF under static operation condition was of the order of $10^{-7}$. Table 5 gives the calculated estimation error based on the error curves shown in Figures 13 and 15a,b.

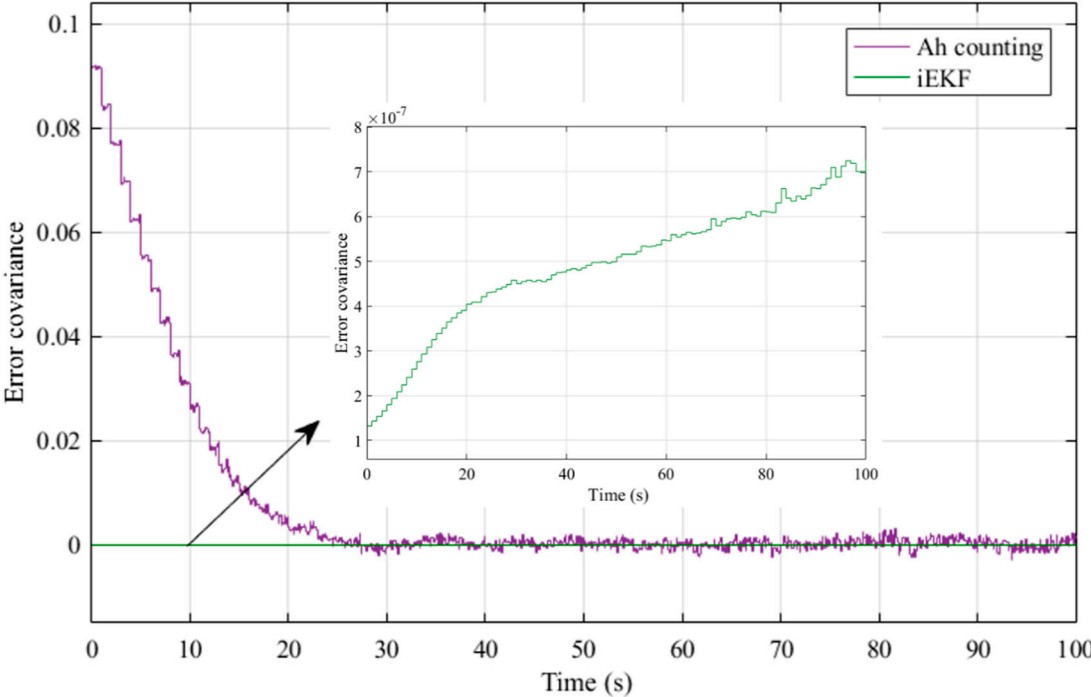

**Figure 13.** Comparison of error covariance between iEKF model and Ah counting model under static operation conditions.

**Table 5.** The SoC estimation error based on the iEKF method under static/dynamic condition.

| Operation Condition | Maximum Error (%) | Average Error (%) | Relative Error (%) |
|---|---|---|---|
| Static | 2.39 | 1.43 | 1.20 |
| Dynamic | 6.76 | 3.94 | 2.15 |

As shown in Figure 12, the estimator based on iEKF shows high degree of agreement tracking the change in SoC. Comparing the curves of estimation error shown in Figures 13 and 15, the iEKF algorithm obtains better noise filtering performance. It should be pointed out that the dynamic operation condition shown in Figure 14 causes more fluctuation in the estimation error curve than in static operation condition. From the comparison results shown in Table 5, the estimation errors increase under dynamic operation conditions. This means the estimation difficulty will become more challenging under dynamic operation conditions.

Since the estimator is sensitive to different operation conditions, the quality of the estimator cannot be fairly compared with different approaches. Several factors such as the battery model, the battery types and the experimental methods for the offline parameter identification affect the performance of the estimator. For example, the estimator based on EKF with the Thevenin model reported an absolute mean error of 4.42% under dynamic operation conditions [49]. Tested results of SoC estimation showed a relative error of 1.5% when using an estimator based on extended fractional KF with the fractional order PNGV model [50]. The SoC estimators based on different battery models using EKF and dual EKF methods have been compared in [42]. Plett [45] built the battery models based on a group of Pulsed-current test and adopted the UDDS dynamic test on LiPB battery, which shown estimation error of the battery models varying from 1% to 6.5%. The iEKF proposed in this paper shows a relative error of 1.2% and 2.15% under static and dynamic conditions, respectively. The SoC estimation based on the improved-EKF model discussed in this paper shows good accuracy and the method itself has less complexity compared to other well-established methods such as the EKF based on the Thevenin or PNGV models.

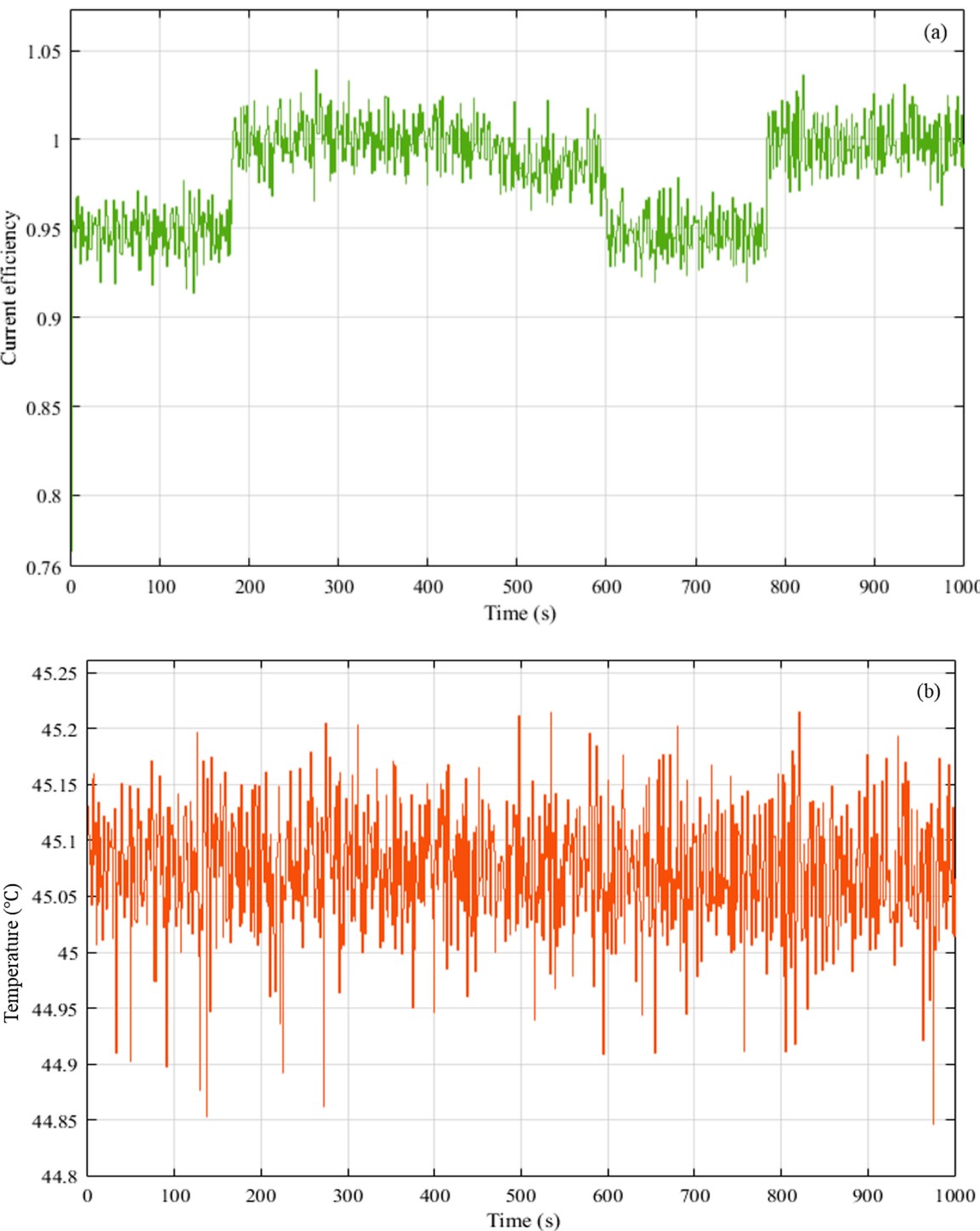

**Figure 14.** MATLAB inputs of (**a**) The input current efficiency and (**b**) The input temperature for the SOC estimation under dynamic operating conditions.

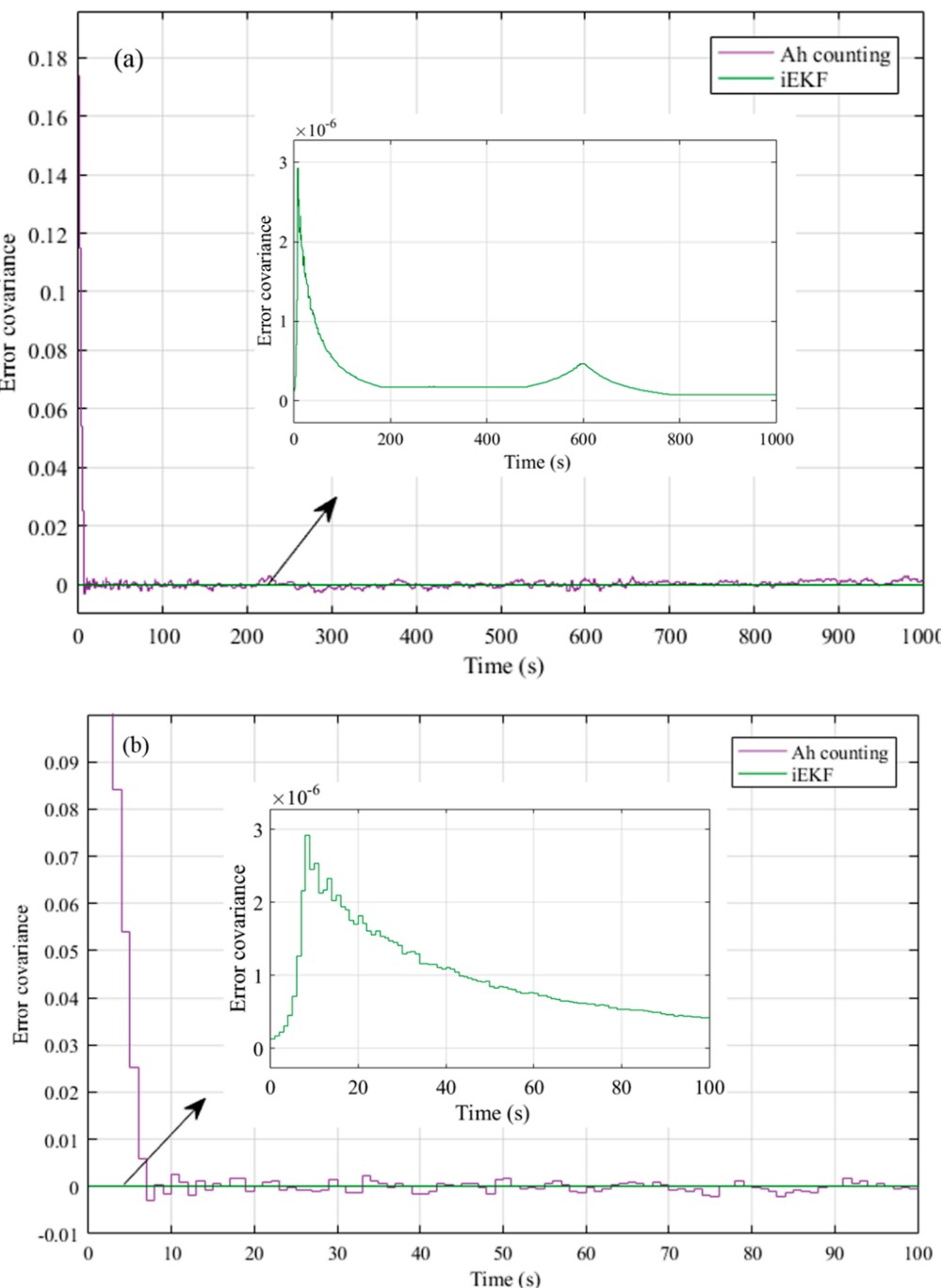

**Figure 15.** Comparison of error covariance between iEKF and Ah models under dynamic operating conditions: (**a**) over a time of 1000 s (**b**) from 0 to 100 s. The inset in both the figures shows the magnified plot of the error covariance of the iEKF model only. Note the extremely small value of error covariance of the iEKF model.

We have also carried out simulations using our iEKF model but with different initial SoCs of 20% and 50% and were able to achieve similar accuracy as that of the SOC value of 70%: relative error under static and dynamic operating condition with initial SoC of 20% and 50% are less than 2%. The relative errors of the SOC estimation for the 20%, 50% and 70% initial guess are compared in Table 6. The results shown in Table 6 further strengthen the credibility of our iEKF model.

**Table 6.** The comparison of relative error (%) between SoC estimation based on the iEKF and Ah counting methods under static/dynamic condition.

| Estimation Error / Operation | Initial SoC of 20% | | Initial SoC of 50% | | Initial SoC of 70% | |
|---|---|---|---|---|---|---|
| | Ah Counting | iEKF | Ah Counting | iEKF | Ah Counting | iEKF |
| Static | 14.9 | 0.8 | 15.3 | 1.0 | 15.0 | 1.2 |
| Dynamic | 17.7 | 1.7 | 18.5 | 2.0 | 19.7 | 2.1 |

## 6. Conclusions

A composite battery model using iEKF has been proposed in this paper. Four groups of real experiments and parameter identifications were conducted to build a reliable battery model for the achieving a credible estimator. A composite battery model was built using offline parameter identification using MATLAB. Based on this composite battery model, a mathematical model of the iEKF was built. OCV method provided the initial estimation inaccuracy and the Ah method provided a rough estimate of the SoC. The cumulative error of SoC estimation in Ah method was precisely corrected by the EKF algorithm. The iEKF algorithm provides successful simulation results for accurate SoC estimation under both static and dynamic operation conditions. The iEKF algorithm shows great advantage in the estimation accuracy while being less complex than other methods. In terms of the simplicity and feasibility, the iEKF is an excellent candidate for BMS implementation to promote the battery performance.

**Author Contributions:** Data curation, N.D. and J.C.; Formal analysis, N.D.; Investigation, N.D.; Methodology, N.D.; Project administration, T.T.L.; Resources, N.D.; Software, N.D.; Supervision, K.P. and T.T.L.; Validation, N.D.; Writing—original draft, N.D.; Writing—review & editing, N.D. and K.P.

**Funding:** This research received no external funding.

**Conflicts of Interest:** The authors declare no conflict of interest.

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
