# Peer review of "State of Charge Estimation of a Composite Lithium-Based Battery Model Based on an Improved Extended Kalman Filter Algorithm"

_inventions, doi:10.3390/inventions4040066_

Round 1
Reviewer 1 Report
This paper gives an SoC estimation method based on a proppsed battery model and EKF. It is a considerable work, but EKF is already used in SoC estimation with other models, the proposed battery model combined with EKF gives good performances but msut be validated in other temperatures like other model. Some improvements have to be add to the article to be published.
- In introduction “Poor bilobliophgy,” only [1] dating from 2013.
- In the introduction: don’t mention how iEKF differ from EKF , UKF , AUKF… ? and why we should use it instead! No critical review is done
- L83 : it seems that the Current is missing in eq2.
- In figure1 : the conception is not really clear, for “the intial parameters identification” and “ online SoC estimation” we can’t understand the relation with the inputs and outputs, please make it more clear !
- The composite model you propose is not well justified, Eq6 how it was demonstrated
( I think a Thevenin model of battery should be detailed , or a reference may be given) , Also in Eq7: the output equation given is not justified , how have you combined the eq3, eq4 and eq5 to have this relation ? a mathematic demarche should give to eliminate OCV(E0)
- 143: the battery used is 206 Ah and 3.2V, is it a single cell or a serial pack, if it is the case please notice.
- The section 3.4 the OCV test is detailed, the purpose of this test is not clear since that the OCV(E0) is deleted from the model you proposed in Eq7? for what parameter the SOC test is used here?
- It would be interesting to see figures that illustrates the variation of ni et nT , in fact the experimental points could be shown and interpolated .
- In Figure.7 , the caption is not precise , also the what does represents “the number of calculation” ? I think that samples should represents the time of charge or discharge
- In section 4.1, there is a lack of references, [23,24] should be mentioned in the correct place in the text not there.
- For the piecewise OCV-SoC relationship, it will be more appropriate to give a figure of the 7 regions you consider of OCV in addition to the equations presented in eq(19).
- In iEKF, you use a combination between a battery model, an OCV-SoC piecewise curve and EKF, this method is well known and used in many works, don’t see why you call it improved EKF, please notice clearly the novelty of you proposition.
- When you are using EKF in your model please give more details: give the details of the application of (6) and (7). Actually it is not clear where you have utilized the (19) the piecewise OCV int the EKF, known that you have eliminate OCV in the model in your composite battery model in (7).
- Xk represents SoC, this may cause an algebraic error, it is better to a state vector Xk composed at least of 2 elements . how did you outcome this problem.
- L341: “Ak-1 and Ck are defined in Step 2 by comparing (6), (7) and (10). The SoC0 is calculated according to the previous remaining battery and current OCV”, this description is ambiguous, OCV is a voltage not a current! And what does it mean the remaining battery? Also the term comparing here is not appropriate I think you mean using or applying?
- L344: “Generally, the data sampling period is decided by the charging/discharging rate and the working conditions”, in your case what sampling period have you used ?
- In figure 13, you compare your SoC estimation based on EKF and your composite model with SoC model. From where you have The SoC model or the real SoC. Obviously it is not the Ah-counting since you have illustrated it in the same figure but you have not mention from where do you have this real SoC!!
- For the initial guess of SoC=70%, is it chosen arbitrary or based on OCV-SoC! please make it clear. In case it is not arbitrary, you may use random initial SoCs to evaluate the convergence of your method.
- To validate the proposed model, you have to apply it in a large marge of temperatures, if it is possible please add the performances at some temperatures other than 25°C.
- This paper does not represent a comparison of the performances (errors, convergences time ...) with other existing method in literatures that uses the same principles. A comparative table will be appreciated.
Malicious
- Figure2: to 4: the titles sizes of figures are too big!
- Figure 5: the (a), (b) and (c) are not mentioned under the subfigures.
- Table 4: Vbefore and Vafter are not explained in the text!
- In figure.7,8, the curves cannot be seen except the red curve, please use different clear colors.
- Captions of figure7 and 8 are not precise for example “linear fitting simulation” of what? , (same for Figure 13 , the simulation results of what ?)
- Legends in Figure 13,14 ,15 are too small and blur, please make it clear
- Please improve the resolution of all simulation figures (PS: don’t use capture)
Author Response
We appreciate all your work and suggestions.
For easier reading, you may use the copied contents here from the response letter:
Response to Reviewer’s Comments
MDPI Inventions
Paper Number: inventions-560218
Paper Title: State of charge estimation of a composite lithium-based battery model based on an improved extended Kalman filter algorithm
Note: Responses are marked with ‘>>>’ and in blue fonts. All the changes in the revised paper in response to reviewer’s comments are shown in red fonts.
Response to Reviewers:
Reviewer 1:
This paper gives an SoC estimation method based on a proposed battery model and EKF. It is a considerable work, but EKF is already used in SoC estimation with other models, the proposed battery model combined with EKF gives good performances but must be validated in other temperatures like other model. Some improvements have to be added to the article to be published.
>>> We would like to thank the reviewers for the positive comments and great suggestions to make it high quality paper. We have carefully read the all comments from the reviewers and explain in detail the changes that we have made for each suggestion. Additionally, we will offer explanations at relevant places to eliminate any doubts and confusion.
1) In introduction “Poor bilobliophgy,” only [1] dating from 2013.
>>> We have re-examined the references in the Introduction section and added more recent and relevant references (reference [1] to [23] from 2010 to 2019) to provide more clarity. The changes are shown in lines 32-35, 37-39, 55-58, 69-70, 73-76 and 84-85 of the revised manuscript.
2) In the introduction: don’t mention how iEKF differ from EKF, UKF, AUKF…? and why we should use it instead! No critical review is done.
>>> We have listed the related SoC estimation works based on different Kalman Filter techniques (EKF, UKF, AUKF) in references [13] to [16]. We have also mentioned how iEKF method differs from other techniques. The changes are shown in lines 82-94 of the revised manuscript.
3) L83 : it seems that the Current is missing in eq2.
>>> The equation 1 is re-written as , where CQ is the remaining capacity and CI is the rated capacity. To further avoid the confusion in equation 2, is changed to , which includes the main factors of charge/discharge rate and temperature . Please see lines 100-102 and 128-131 of the revised manuscript.
4) In figure1: the conception is not really clear, for “the initial parameters identification” and “online SoC estimation” we can’t understand the relation with the inputs and outputs, please make it more clear !
>>>As suggested by reviewer, figure 1 is replaced with a better figure to give a clearer conception. The caption for the figure is also changed. A brief description has also been added to explain the relation between the input and output. Please see Figure 1 and lines 110-122 of the revised manuscript.
5) The composite model you propose is not well justified, Eq6 how it was demonstrated
( I think a Thevenin model of battery should be detailed , or a reference may be given) , Also in Eq7: the output equation given is not justified , how have you combined the eq3, eq4 and eq5 to have this relation ? a mathematic demarche should give to eliminate OCV(E0)
>>> The composite model adopted in this research is a novelty where three different electrochemical models of the battery (equations 3, 4, and 5) are combined into one. The K0 in equation 7 is now clearly defined. More explanation is also added to emphasize that K0 in our model is the same as E0 of other models with the difference that K0 is calculated OCV while E0 is directly measured OCV. Please see lines 150-164 of the revised manuscript.
6) L143: the battery used is 206 Ah and 3.2V, is it a single cell or a serial pack, if it is the case please notice.
>>>The battery is single cell. It is now clearly shown in the revised manuscript (lines 172-173).
7) The section 3.4 the OCV test is detailed, the purpose of this test is not clear since that the OCV(E0) is deleted from the model you proposed in Eq7? for what parameter the SOC test is used here?
>>> We agree with the reviewer that the OCV test in section 3.4 is used to build the relationship between the OCV (K0) and the SoC (xk). With a clearer definition of K0, we believe there is no further confusion here.
8) It would be interesting to see figures that illustrates the variation of ni et nT, in fact the experimental points could be shown and interpolated.
>>> and are obtained through the Polyfit Function in Matlab based on the experimental data. We have shown the function results in equations (8) and (9) and feel that there is no necessity for the interpolation of data.
9) In Figure.7, the caption is not precise, also the what does represents “the number of calculation”? I think that samples should represents the time of charge or discharge.
>>>In response to reviewer 2, we have decided to remove these two figures and give the converged parameter values after 1400 iterations.
10) In section 4.1, there is a lack of references, [23,24] should be mentioned in the correct place in the text not there.
>>> We have added references in specific places in section 4.1.
11) For the piecewise OCV-SoC relationship, it will be more appropriate to give a figure of the 7 regions you consider of OCV in addition to the equations presented in eq(19).
>>> We believe that adding a figure for equation (19) does not add any more clarity than what the equations show now. Hence we have decided not to add the figure.
12) In iEKF, you use a combination between a battery model, an OCV-SoC piecewise curve and EKF, this method is well known and used in many works, don’t see why you call it improved EKF, please notice clearly the novelty of you proposition.
>>> To clarify the novelty of the iEKF and the contributions of this paper, we have summarized and added explanations in the last paragraph of introduction section (lines 82-94). As the reviewer pointed out, KF based methods for SoC estimation in other papers generally combine the OCV approach and the KF/EKF algorithms. The OCV-SoC function in our work is providing an initial value for SOC estimation. Then we build the Ah counting module to online identify the parameters of the battery model. Subsequently, we use the EKF to correct the errors in the Ah counting. We have called this overall technique as the iEKF method and shown that it provides more accurate estimation of SoC. This is now clearly indicated in the revised manuscript (lines 82-94).
13) When you are using EKF in your model please give more details: give the details of the application of (6) and (7). Actually it is not clear where you have utilized the (19) the piecewise OCV int the EKF, known that you have eliminate OCV in the model in your composite battery model in (7).
>>> We have made clear distinction between E0 and K0 at the end of section 2. As a result, now the utilization of equation (19) should be clear.
14) Xk represents SoC, this may cause an algebraic error, it is better to a state vector Xk composed at least of 2 elements. how did you outcome this problem.
>>> As the reviewer mentioned, the Algebraic loop issue cannot be avoided when using the Simulink to build this iEKF model. We adopted the Delay unit in Simulink, which uses the Delay unit as a separate feedback. The Delay unit is not a pass-through module, the algebraic error can be ignored. The disadvantage is that the system takes longer time to calculate during dynamic simulation. In terms of the xk, please see the change we made in lines 133-134 of the revised manuscript.
15) L341: “Ak-1 and Ck are defined in Step 2 by comparing (6), (7) and (10). The SoC0 is calculated according to the previous remaining battery and current OCV”, this description is ambiguous, OCV is a voltage not a current! And what does it mean the remaining battery? Also the term comparing here is not appropriate I think you mean using or applying?
>>> A clearer explanation is now provided. Please see lines 364-366 of the revised manuscript.
16) L344: “Generally, the data sampling period is decided by the charging/discharging rate and the working conditions”, in your case what sampling period have you used?
>>> >>> The ‘data sampling period’ is the reciprocal of sampling frequency. The sampling frequency is set to a value of two and half times the bandwidth of the sampled signal. This is now indicated in lines 366-368.
17) In figure 13, you compare your SoC estimation based on EKF and your composite model with SoC model. From where you have The SoC model or the real SoC. Obviously it is not the Ah-counting since you have illustrated it in the same figure but you have not mention from where do you have this real SoC!!
>>> Explanation is now provided in lines 376-379. Please note that the figure number has changed to Fig. 11.
18) For the initial guess of SoC=70%, is it chosen arbitrary or based on OCV-SoC! please make it clear. In case it is not arbitrary, you may use random initial SoCs to evaluate the convergence of your method.
>>>The initial guess of SoC = 70% is the same as the most of the SoC estimation research (lines 373-374). We have added the related references.
19) To validate the proposed model, you have to apply it in a large marge of temperatures, if it is possible please add the performances at some temperatures other than 25°C.
>>> Figure 15b was obtained using a temperature of and is a dynamic temperature while the static temperature setting was .
20) This paper does not represent a comparison of the performances (errors, convergences time ...) with other existing method in literatures that uses the same principles. A comparative table will be appreciated.
>>> Figure 11 shows the comparison between the standard SoC, Ah counting method and iEKF. It is not possible to directly compare our results with the existing results due to the differences in the methods and model.
Malicious
- Figure2: to 4: the titles sizes of figures are too big!
>>>All replaced by a better one as the reviewer suggested.
- Figure 5: the (a), (b) and (c) are not mentioned under the subfigures.
>>>They are shown in the subfigures.
- Table 4: Vbefore and Vafter are not explained in the text!
>>>The explanation has been added in lines 230-231 of the revised manuscript.
- In figure.7,8, the curves cannot be seen except the red curve, please use different clear colors.
>>> The figures 7 and 8 are removed.
- Captions of figure7 and 8 are not precise for example “linear fitting simulation” of what? , (same for Figure 13 , the simulation results of what ?)
>>>The figures 7 and 8 are removed.
- Legends in Figure 13,14 ,15 are too small and blur, please make it clear
>>>The figures are replaced by clearer resolution (now shown in Figure 11, 12, and 13).
- Please improve the resolution of all simulation figures (PS: don’t use capture)
>>>All simulation figures are replaced by exported .png documents from MATLAB.
Reviewer 2 Report
I felt that this paper was potentially interesting, but there were some serious issues I would like to see addressed:
The paper does not take into account the fact that most EKF-based models are inherently based on a combination of OCV with a model. Plett's papers (2004) are an excellent example of this. There is nothing particularly novel about doing this - it's standard practice. The composite battery model is potentially novel, but it's very poorly justified. It appears to have been simply thrown together. A more rigorous justification explaining why it is scientifically valid to combine the terms is necessary and helpful. If I understand Figure 7, correctly the parameter values do not seem to be converging to anything sensible. This is an argument against the model. Figure 13 seems to be meaningless - and cannot be correct. The excitation tests are much more benign than would be seen in a typical application. The relationship between x and SoC is not intuitive and needs more explanation. The literature is insufficiently thorough, and misses out some key well known texts (like the Plett papers).
Author Response
We appreciate all your work and suggestions.
For easier reading, you may use the copied contents here from the response letter:
Response to Reviewer’s Comments
MDPI Inventions
Paper Number: inventions-560218
Paper Title: State of charge estimation of a composite lithium-based battery model based on an improved extended Kalman filter algorithm
Note: Responses are marked with ‘>>>’ and in blue fonts. All the changes in the revised paper in response to reviewer’s comments are shown in red fonts.
Response to Reviewers:
Reviewer 2:
I felt that this paper was potentially interesting, but there were some serious issues I would like to see addressed:
The paper does not take into account the fact that most EKF-based models are inherently based on a combination of OCV with a model. Plett's papers (2004) are an excellent example of this. There is nothing particularly novel about doing this - it's standard practice.
>>>Thank you so much for suggesting Plett's papers. We read and compared our work and the works in Plett's papers. In the part II and III of Plett's papers, listed five types of different models and each of them is used conjunction with OCV and KF/EKF method. In terms of the novel issue of our work, this is similar to the queries by reviewer 1 (Points 2 and 12). We have provided the necessary explanations there and also made changes in the manuscript (lines 82-94). We have also added Plett’s papers in our references.
The composite battery model is potentially novel, but it's very poorly justified. It appears to have been simply thrown together. A more rigorous justification explaining why it is scientifically valid to combine the terms is necessary and helpful.
>>>This is similar to the queries by reviewer 1 (Points 5 and 7). Please refer to our responses and changes made for reviewer 1 (lines150-164).
If I understand Figure 7, correctly the parameter values do not seem to be converging to anything sensible. This is an argument against the model.
>>>The parameters values appeared to be converging after about 1400 iterations. The final values of the parameters are shown in lines 259-263. The accuracy in the estimation of parameters was high, of the order of ±0.3%. This is also indicated in the manuscript. The two relevant figures are removed to prevent any confusion.
Figure 13 seems to be meaningless - and cannot be correct. The excitation tests are much more benign than would be seen in a typical application.
>>>The figure has been redrawn.
The relationship between x and SoC is not intuitive and needs more explanation.
>>>This has been answered in relation to questions by reviewer 1 (Point 14).
The literature is insufficiently thorough, and misses out some key well known texts (like the Plett papers).
>>>The reference list has been updated and includes papers by Plett.
Reviewer 3 Report
The topic of the batteries for electric vehicle (EV) applications today is of much interest, and then all the researches directed towards these issues are particularly important. In this paper, a method for SoC estimation under both static and dynamic operating conditions has been implemented by using a combination of battery model and iEKF algorithm. The results are useful to describe in general the behavior that batteries can have under real driving scenario and to estimate in particular the batteries SOC. Interesting experiments on LiFePO4 Li-ion battery have been performed to obtain real data useful for model validation. A methodology to build an accurate models able to estimate SOC was proposed. This methodology was demonstrated using iron-phosphate battery data, but the authors highlighted as this general methodology can be universally and usefully applied to any type of battery for a BMS control system development. This work highlights, by a simulation, the control strategy for the EV. It can be considered of interest to the scientific community, it provides own contribution to science and technology pushing the researches towards the electric car applications. It is quite well done and the results seem to be coherent, demonstrating clearly and linearly the thesis of the paper, therefore there are not relevant changes to do and even if it isn’t particularly original it could be published.
Round 2
Reviewer 1 Report
I would like to thank the authors for the improvements made in the paper, the presentation is now more appropriate and most of ambiguities are eliminated.
But still need some clarifications:
I still believe that Eq 2 is not well written. Actually by definition the quantity of charge of a battery where I is the current across the battery . in your case I don’t see this! You have how have you eliminated you are integrating ɳ that we cannot see the variation with time.
For the initial guess of SOC it will be interesting to show the convergence of the proposed method beginning from different initial SoC (0%, 20%, 50%, 70%, 100%) as examples, and this to prove the robustness of the algorithm against false initial SOC determination and cumulative errors.
It is regrettable that you cannot compare you IEKF to standard EKF or UKF to prove the effectiveness of your proposed approach. This will be a considerable achievement.

Author Response
We would like to thanks for all the considerations and suggestions from the reviewers and editor. We have tried to eliminate all the doubts and achieve more expectations to make a high-quality paper.
>>>Thanks for pointing this typographical mistake. We have corrected the expression in equation (2). The changes are shown in lines 128-130 of the revised manuscript (round 2).
>>>We have added a table (Table 6) comparing the different initial SOCs. Please see lines 444-448 of the revised manuscript (round 2).
Reviewer 2 Report
Despite the authors' modification, I feel that the most serious issues identified before remain.
The paper does not take into account the fact that most EKF-based models are inherently based on a combination of OCV with a model. Although more has been added to the introduction, it remains the case that there is nothing particularly novel about doing this. The paper does now cite Plett's papers (2004). There is nothing particularly novel about doing this - it's standard practice.
I feel that the composite battery model, which could potentially be novel, remains very very poorly justified. I feel that it appears to have been simply thrown together from parts of other models. I would still like to see a more rigorous justification explaining why it is scientifically valid to combine the terms is necessary and helpful. The authors' revisions do little if anything to address this fundamental concern. I understand where the parts come from - but the authors need to show that combining the models in this way is scientifically valid.
The authors' response to my concern regarding Figure 7 has been to delete it - this seems to me to be a bad practice. A better response would have been to demonstrate convergence properly - it seems wrong to simply delete a figure when it does not support the paper's argument without providing evidence that the 'statement it appears to contradict is true. If the values really do converge, this should be easy to show by extending the timeline further into the future.
Figure 14 (which was Figure 13) still does not show what the caption says it does - it needs more clarity.
I still feel that the excitation tests are much more benign than would be seen in a typical application. I did not see a response to this comment from the authors.
Author Response
We would like to thanks for all the considerations and suggestions from the reviewers and editor. We have tried to eliminate all the doubts and achieve more expectations to make a high-quality paper.
Despite the authors' modification, I feel that the most serious issues identified before remain.
• The paper does not take into account the fact that most EKF-based models are inherently based on a combination of OCV with a model. Although more has been added to the introduction, it remains the case that there is nothing particularly novel about doing this. The paper does now cite Plett's papers (2004). There is nothing particularly novel about doing this - it's standard practice.
>>>We had already provided some explanation of the novelty of our method compared to other methods (lines 82-94). We are not sure what specific novelty the reviewer is expecting here.
I feel that the composite battery model, which could potentially be novel, remains very very poorly justified. I feel that it appears to have been simply thrown together from parts of other models. I would still like to see a more rigorous justification explaining why it is scientifically valid to combine the terms is necessary and helpful. The authors' revisions do little if anything to address this fundamental concern. I understand where the parts come from - but the authors need to show that combining the models in this way is scientifically valid.
>>>We have reviewed several papers, discussed and compared different battery models and have given a summary in lines 159-178 of the revised manuscript (round 2). A good SoC estimator will predominantly depend on an accuracy of the battery model.
The authors' response to my concern regarding Figure 7 has been to delete it - this seems to me to be a bad practice. A better response would have been to demonstrate convergence properly - it seems wrong to simply delete a figure when it does not support the paper's argument without providing evidence that the 'statement it appears to contradict is true. If the values really do converge, this should be easy to show by extending the timeline further into the future.
>>>We have done longer simulation and show the results in figure 7 clearly showing convergence of the parameters. Please see lines 273-280 of the revised manuscript (round 2).
Figure 14 (which was Figure 13) still does not show what the caption says it does - it needs more clarity.
>>>This figure is redrawn and give a clearer and more specific caption. Please see figure 15 and lines 402-404 of the revised manuscript (round 2).
I still feel that the excitation tests are much more benign than would be seen in a typical application. I did not see a response to this comment from the authors.
>>>We are confused about the “excitation test” mentioned by the reviewer and are unable to understand which part the reviewer is referring to. We will appreciate more detailed clarification from the reviewer.
Round 3
Reviewer 2 Report
I am not satisfied that the authors have not satisfactorily addressed the concerns raised at previous review stages.